# An In-Depth Statistical Analysis of the TARC Parameter to Evaluate the Real Impact of Random Phases in MIMO Antennas

**DOI:** 10.3390/s25134171

**Published:** 2025-07-04

**Authors:** Angel Perez-Miguel, Hildeberto Jardon-Aguilar, Jose Alfredo Tirado-Mendez, Ricardo Gomez-Villanueva, Ruben Flores-Leal, Erik Fritz-Andrade

**Affiliations:** 1Telecommunications Section, CINVESTAV-IPN, Av IPN 2508, San Pedro Zacatenco, Mexico City 07360, Mexico; aperezmi@cinvestav.mx (A.P.-M.); hjardon@cinvestav.mx (H.J.-A.); rgomez@cinvestav.mx (R.G.-V.); rfleal@cinvestav.mx (R.F.-L.); 2Electrical Engineering Department, SEPI-ESIME-Zacatenco, Instituto Politécnico Nacional, Av IPN S/N, Edif. 5, Mexico City 07360, Mexico; 3Asociación de Normalización y Certificación (ANCE), Eje Central Lázaro Cárdenas 869, Nueva Industrial Vallejo, Mexico City 07700, Mexico; erik.fritz@ance.org.mx

**Keywords:** total active reflection coefficient, MIMO antenna array, PIFA antenna, envelope correlation coefficient, capacity loss, MIMO communication, low mutual coupling, random TARC, high MIMO efficiency

## Abstract

A detailed statistical analysis of the total active reflection coefficient (TARC) is carried out in this paper for three 4-port MIMO antennas featuring different levels of isolation across its ports. This analysis is very useful to determine the most likely performance of a MIMO antenna in a real communications scenario. The TARC parameter is commonly evaluated for only several combinations of the random phase with which a signal reaches every input port of a MIMO antenna. By contrast, we have evaluated a million combinations to obtain the probability density function of the TARC, using frequency as its parameter. In this way, an expected value of the TARC is obtained for each frequency, as well as a confidence interval ($\Delta CI_{\mathrm{TARC}}$) where the TARC values occur with 90% probability. Additionally, we have introduced the term “TARC shadow”, a visual representation of the TARC as a function of the frequency where the probability function is projected into this 2D graphic with different colors to identify the most likely values of the TARC. To demonstrate these concepts, a full TARC evaluation was performed for three 4-port MIMO antennas with increasing isolation of 12.9 dB, 25.4 dB, and 37 dB between elements, and different values of the $S_{nn}$ and $S_{nm}$ parameters, with *n* and m= 1 to 4. From this study, the importance of the isolation among ports and its comparison with the return losses becomes evident in achieving a MIMO antenna array insensitive to random phase variations occurring in the communication channel.

## 1. Introduction

A notable trend in wireless communications is the continuous pursuit of higher data rates accompanied by enhanced reliability. The advent of 5G witnessed significant developments in multiple-input-multiple-output (MIMO) antennas, which are now the basis of the forthcoming 6G evolution [1,2]. MIMO antennas offer several advantages, including improved link reliability, increased channel capacity, reduced co-channel interference, higher data rates, and enhanced spectral efficiency [3,4,5]. To better appreciate the effect of the MIMO antennas in the context of 4G to 6G, examining the specific mechanisms through which these advantages are achieved is crucial.

Several figures of merit are commonly analyzed to evaluate the advantages of MIMO antennas, including envelope correlation coefficient (ECC), diversity gain (DG), capacity loss ($C_{\mathrm{loss}}$), multiplexing efficiency, and mean effective gain (MEG). These parameters provide perspectives into the performance and capabilities of the MIMO antennas. In many cases, models of these parameters are simplified to facilitate their evaluation and measurement. However, it is important to emphasize that MIMO systems require highly uncorrelated multipaths to increase the channel capacity, and the magnitudes, phases, and directions of the signals arriving to the MIMO antenna are inherently random. Therefore, it is essential to consider their random nature to obtain a more accurate analysis of the efficiency of MIMO systems. This means that the main antenna parameters should include probabilistic behavior across the entire bandwidth of interest.

The concept of TARC was introduced in [6,7] to characterize the bandwidth and radiation performance of N-ports antennas considering the total incident power and the total reflected power. The TARC is assessed by the square root of the ratio of these signals, and it is equal to one when the total incident power is reflected and zero if it is completely accepted. Certainly, the information of the incident and reflected power amplitudes is contained in the S-parameters of the N-ports system. However, the phases of the incident and reflected signals must also be considered in the TARC calculation as reported in [8,9,10].

The study outlined in [11] highlights the importance of the random phases in a MIMO antenna. Under certain conditions, the random phases can cause significant variations in radiation efficiency, affecting the overall performance of the antenna system. For example, in a beamforming antenna array the phase of the signals is critical to control the direction of the wave front. And if the antenna array changes its bandwidth depending on the phase of the signals, the result can be a reduction in the efficiency of the whole system. In these cases, the TARC acquires great relevance since it determines how an antenna array behaves for all the combinations of phases of the signals. Nevertheless, the degradation of the array performance, using the TARC parameter, has not been sufficiently well analyzed so far [12,13,14,15].

In this paper, the statistical behavior of the TARC of a 4-port MIMO antenna under realistic propagation conditions is analyzed in depth. For this purpose, the TARC is evaluated considering one million combinations of the input phases that arrive at the four ports of the MIMO antenna. This number of combinations was obtained using a function programmed in MATLAB^®^ R2024b software, that generates random phases, according to a uniform probability distribution. Then, the TARC values are obtained applying these combinations of phases and the measured S-parameters of the MIMO antenna. The result is one million TARC values that follow a particular probability distribution function (PDF) from which the mean and dispersion values are calculated. The PDF dispersion value that was used in this study is the confidence interval ($\Delta CI_{\mathrm{TARC}}$) that covers the TARC values that occur with 90% probability.

The previous procedure is carried out for every frequency measured around the central frequency of the MIMO antenna. In this case, the evaluation is performed for 401 frequencies in the interval from 5.6 GHz to 6.0 GHz. In order to gain a closer physical insight into this massive amount of data, the variables of frequency, TARC, and probability are used to create a 3D representation of the probability density of the TARC that allows for a comprehensive data analysis of its expected value and the bandwidth in the MIMO antenna. However, data analysis using 3D graphs is not so easy to visualize. For this reason, and to make a straightforward comparison among bandwidth and efficiency in MIMO antennas, using the TARC data, a 2D representation of the entire dataset called “TARC shadow” was created. In this graphic of the TARC as a function of frequency, a color is assigned to every single point in the 2D graphic, depending on the probability that this TARC punctual value occurs at a given frequency. In this way, the red points indicate the maximum peaks of probability, whereas the blue points specify the minimum values of TARC probability. Along with this resultant graphic resembling a shadow, a curve with the mean TARC value for each frequency is plotted to give an overall idea of the TARC behavior of a MIMO antenna under real operative conditions. Due to the TARC being an inherent parameter that measures the total bandwidth of the MIMO antenna, the TARC shadow can be applied to any topology MIMO array regardless of the type of radiator.

## 2. Proposed MIMO Antennas with Several Levels Inter-Port Isolation

The considered MIMO antenna for the TARC analysis consists of an array of four planar inverted F-antenna (PIFA) elements. Figure 1a is a 3D view of each PIFA element where a feeding terminal, the current return plane, and the current return connection are identified. The feeding point is located near the center of the top plate of the PIFA, while the current return connection is placed in the upper right corner of the top plate as shown if Figure 1a. Five AWG 26 copper vias establish the returning connection, extending from the PIFA’s top plate through the current return plane. While the substrate is not visually represented for enhanced clarity of the via placement, its specifications are a thickness of h=3.175, and material Taconic TLY-5, possessing a relative permittivity of 2.2.

The single PIFA element was first designed for a central frequency of 5.8 GHz and was then optimized using the CST Studio Suite^®^ Release Version 2019.01 software [16]. The final dimensions of the single PIFA element, according to Figure 1b, were the following: $w_{1}=8$ mm, $l_{1}=21$ mm, $l_{2}=17.5$ mm, $l_{s}=3.5$ mm, $w_{s}=1.4$ mm, and $l_{\mathrm{feed}}=1.5$ mm. With these dimensions, the $S_{11}$ parameter at 5.8 GHz was −35 dB, and the bandwidth was 200 MHz or 3.44% of fractional bandwidth.

The arrays based on the PIFA of Figure 1 are illustrated in Figure 2, where the element’s separation is set to $\lambda_{0}/2$. These arrays are comprised of four elements and the numbers 1 to 4 in each PIFA element are associated with the indexes of the S-parameters. To reduce the mutual coupling among the closest elements (horizontal and vertical neighbors), the polarization diversity technique is applied, while the diagonal elements maintain the same polarization. The MIMO PIFAs of Figure 2 are symmetrical arrays, where the conditions of a reciprocal network are satisfied ($S_{nm}=S_{mn}$), and additionally, the S-parameters of adjacent elements are approximately the same (e.g., $S_{12}\approx S_{13}$). The complete area of the MIMO antennas occupy a rectangular area with dimensions of L=W=60 mm.

Since the TARC is mainly affected by mutual coupling [11], it is necessary to increase the isolation of elements within a MIMO antenna to improve the TARC parameter. Therefore, this work focuses on the statistical analysis of the TARC for three MIMO antennas with different isolation levels to determine the TARC performance when the coupling is reduced. This statistical analysis also proves how bandwidth and efficiency are affected by the random phases at the input ports of each MIMO antenna. Furthermore, the MIMO antennas are studied under the conditions of $|S_{nm}\left|\;>\;\right|S_{nn}|$, $|S_{nm}\left|\;\approx \;\right|S_{nn}|$, and $|S_{nm}\left|\;<\;\right|S_{nn}|$, which are also important for the TARC values within the antenna bandwidth. These conditions on the S-parameters were obtained for the three MIMO antennas with minor geometry modifications in the return plane to change the surface currents and therefore the mutual decoupling levels.

The three MIMO antennas of Figure 2 have the same dimensions, but some geometry variants were introduced to progressively enhance the isolation between ports. The first antenna, called MIMO PIFA 1, is depicted in Figure 2a. In this antenna, the current return connection is located at the lower right corner of the PIFA element number 1, as indicated in Figure 2a. This return connection is also indicated as $A^{*}$ in Figure 2b. In the second antenna, designated as MIMO PIFA 2, the current return connection is changed to the top right corner of the PIFA element number 1, as shown in Figure 2b. This simple modification improves the isolation as will be discussed later. Finally, the third antenna, named MIMO PIFA 3, is based on the second antenna, with the addition of a cross-shaped slot in the current return plane that further improves the isolation across its ports. The dimensions of this slot are $w_{r}$ and $L_{r}$, as illustrated in Figure 2c, and are equal to 1 mm and 50 mm, respectively. This Defective Ground Structure (DGS) is based on the MIMO antenna presented in [17], where a parametric study of $w_{r}$ and $L_{r}$ was performed to improve isolation, almost without affecting the $S_{nn}$ parameters, bandwidth, efficiency, and gain of the antenna.

Figure 3 illustrates the current distributions on the return plane of each MIMO antenna when PIFA element 1 is fed, and the other ports are loaded at 50 Ω. These surface currents on the return plane define how the neighboring ports are affected. In Figure 3a it is observed how MIMO PIFA 1 spreads the surface currents from the feeding point toward the array’s center, resulting in moderate coupling with PIFAs 2 and 4. The predominant colors in this figure are green and yellow, indicating a strong mutual coupling. However, by modifying the positions of the current return connection, the main path of the surface currents is directed toward the top of the array, as shown in Figure 3b. This reduces the current density originating from the feeding point to PIFAs 4 and 3. This small change results in an improved current distribution, as seen by the appearance of blue and turquoise areas that indicate a weak coupling. Figure 3c exhibits the current distribution with the implementation of the DGS. Here, the predominant blue color inside PIFAs 3 and 4 indicates an even lower density and, hence, the lowest coupling. This effect is also observed to a lesser extent with PIFA 2, where some blue areas are noted around this element, compared to the density presented in Figure 3b.

To corroborate what the aforementioned current distributions suggest, the S-parameters of the three MIMO antennas were measured. In Figure 4a, the measured mutual coupling at the central frequency of 5.8 GHz between the adjacent elements PIFA 1 and PIFA 2 of the first MIMO antenna is indicated as $S_{12}=-16.2$ dB. On the other hand, the measured mutual coupling of the opposite elements PIFA 1 and PIFA 4, at the same central frequency, is marked in Figure 4a as $S_{14}=-13$ dB. Modifying the current distribution of the second MIMO antenna results in improved isolation where $S_{12}$ and $S_{14}$ drop to −25.9 dB and −27.5 dB, respectively, as indicated in Figure 4b. However, the best isolation is found in the third MIMO antenna that incorporates the DGS and where $S_{12}$ and $S_{14}$ decrease even further to −36.9 dB and −37.1 dB, as can be seen in Figure 4c.

Note that the measured curves of $S_{12}$, $S_{13}$, and $S_{14}$ ($S_{nm}$) have different levels with respect to the measured curves of $S_{11}$, $S_{22}$, $S_{33}$, and $S_{44}$ ($S_{nn}$) for each MIMO antenna. In Figure 4a the level of the $S_{nm}$ curves is higher than that of the $S_{nn}$ curves around the central frequency of 5.8 GHz. However, the levels of the curves of $S_{nm}$ are comparable to the $S_{nn}$ curves at frequencies closer to the central frequency for the second MIMO antenna, as can be observed from Figure 4b. And remarkably, the $S_{nm}$ curves are always below the $S_{nn}$ curves, for all frequencies measured at the third MIMO antenna as shown in Figure 4c. This brings us to three case studies of interest that will be considered later when determining the TARC parameter under the special conditions of $|S_{nm}\left|\;>\;\right|S_{nn}|$, $|S_{nm}\left|\;\approx \;\right|S_{nn}|$, and $|S_{nm}\left|\;<\;\right|S_{nn}|$.

## 3. Statistical TARC Analysis

TARC is an important MIMO parameter that will be analyzed in detail in this section, considering its statistical behavior versus the random phases. The TARC depends on the total incident and reflected signals on the N ports of a MIMO antenna and can be assessed by the following expression [6,9]:(1)$$\mathrm{TARC}=\frac{\sqrt{\sum_{n=1}^{N}{\left|b_{n}\right|}^{2}}}{\sqrt{\sum_{n=1}^{N}{\left|a_{n}\right|}^{2}}}$$
where *N* is the number of ports of the MIMO antenna, and $a_{n}$ and $b_{n}$ are the incident and reflected signals, respectively. The vectors of incident and reflected signals are related by the S-parameters matrix, according to (Equation 2).(2)b=Sa

If the signal $a_{1}$ that arrives at port 1 of the MIMO system is considered as the reference with zero phase, the random phases of all the other incoming signals are related to the incident signal $a_{1}$ by:(3)$$a_{n\neq 1}=a_{1}e^{j\theta_{1,n}}$$
where $\theta_{1,n}=\theta_{1}-\theta_{n}$ is the random phase difference between port 1 and the *n*th port, which may take any value from 0 to 2π. This phase difference $\theta_{1,n}$ is unknown and depends upon the real statistical conditions of the propagation channel. Here, this phase difference is considered as a uniformly distributed random variable (RV), which represents a fair approximation of the real propagation conditions for many cases [18].

If Equation (Equation 1) is developed using expressions (Equation 2) and Equation 3, a compact equation is obtained to evaluate the TARC fo any *N*-port MIMO antenna [8].(4)$$\mathrm{TARC}=\frac{1}{\sqrt{N}}\sqrt{\sum_{n=1}^{N}{\left|S_{n1}+\sum_{m=2}^{N}S_{nm}e^{j\theta_{1,m}}\right|}^{2}}$$

If we take N=4 in Equation (Equation 4), the TARC can be expressed in its extended form as shown in Equation (Equation 5), which allows us to analyze the weight that each random variable has in the evaluation of the TARC. It is also observed in Equation (Equation 5) that variables $S_{11}$, $S_{21}$, $S_{31}$, and $S_{41}$ are deterministic, while the others are random.(5)$$\mathrm{TARC}=\frac{1}{2}\sqrt{\begin{array}{l} {\left|S_{11}+S_{12}e^{j\theta_{1,2}}+S_{13}e^{j\theta_{1,3}}+S_{14}e^{j\theta_{1,4}}\right|}^{2}+\\ {\left|S_{21}+S_{22}e^{j\theta_{1,2}}+S_{23}e^{j\theta_{1,3}}+S_{24}e^{j\theta_{1,4}}\right|}^{2}+\\ {\left|S_{31}+S_{32}e^{j\theta_{1,2}}+S_{33}e^{j\theta_{1,3}}+S_{34}e^{j\theta_{1,4}}\right|}^{2}+\\ {\left|S_{41}+S_{42}e^{j\theta_{1,2}}+S_{43}e^{j\theta_{1,3}}+S_{44}e^{j\theta_{1,4}}\right|}^{2}\;\;\;\;\; \end{array}}$$

In (Equation 5), the reference phase is associated to port 1, so when n=1, $\theta_{1,1}$ is equal to zero. If the phase difference is a random variable, it is clear that TARC is another random variable that depends on the real propagation conditions.

The TARC parameter is usually reported as a family of curves each associated with the $\theta_{1,n}$ phases of the MIMO antenna ports, other than the phase of port 1 which is taken as a reference [19,20,21]. However, the number of phases taken into account to obtain the TARC is very limited and does not give an accurate picture of this parameter. For this reason, in this paper we propose to find accurately the characteristics of the TARC random variable using three vectors of 1,000,000 values for each random variable $\theta_{1,2}$, $\theta_{1,3}$, and $\theta_{1,4}$. These values were generated using a MATLAB^®^ function that randomly picks a number within the range of 0 to 2π, following a uniformly distributed random variable. Selecting in consecutive order three phases of the generated vectors, and using the measured S-parameters in a specific frequency, a total of 1,000,000 values of TARC are calculated for the three MIMO-PIFA antennas. With the information provided by this amount of TARC data, it is possible to develop a complete statistical analysis not limited to a small set of phase combinations. A more comprehensive explanation of this procedure is provided by a flowchart in Appendix A.

Using this algorithm implemented in MATLAB^®^, the probability density functions of the TARC are studied for three cases: $|S_{nm}\left|\;>\;\right|S_{nn}|$, $|S_{nm}\left|\;\approx \;\right|S_{nn}|$, and $|S_{nm}\left|\;<\;\right|S_{nn}|$. The frequencies for the TARC analysis are selected for each MIMO antenna where the best ratio within the bandwidth is obtained from de measured S-parameters. The first selected frequency is 5.82 GHz with a ratio of $|S_{nm}\left|/\right|S_{nn}|=14$ for the first MIMO antenna, the second frequency is 5.78 GHz with a ratio of $|S_{nm}\left|/\right|S_{nn}|=1.0$, measured for the second MIMO antenna, and the third frequency is 5.7 GHz, corresponding to the lower cut-frequency of the third MIMO antenna, with a ratio of $|S_{nn}\left|/\right|S_{nm}|=10$. The probability density functions of the TARC for MIMO PIFAs 1, 2, and 3 are depicted in Figure 5. In each case studied, the mean value E[TARC] is highlighted in red letters on the upper left side that corresponds for the red dashed line within of each histogram. Additionally, a confidence interval ($\Delta CI_{\mathrm{TARC}}$) was obtained where 90% of the possible values of TARC would occur, with the limits indicated by dashed black lines, and the interval $\Delta CI_{\mathrm{TARC}}$ indicated by a green dashed line.

Analyzing the confidence interval of Figure 5a, it can be observed that the MIMO antenna with the highest mutual coupling originates a wide TARC PDF dispersion. In this case, there is a probability of 90% that the TARC varies from −14.6 dB to −6.9 dB. That means an interval of $\Delta CI_{\mathrm{TARC}}$=7.7 dB. On the other hand, Figure 5b reports an interval $\Delta CI_{\mathrm{TARC}}$ even higher of 8.2 dB, where the TARC fluctuates from −25 dB to −16.8 dB. A very different case is MIMO PIFA 3, where the PDF dispersion was significantly reduced, with PDF variations from −11.7 dB to −10.5 dB, which is an interval of $\Delta CI_{\mathrm{TARC}}$=1.2 dB. The dispersion of each MIMO antenna can be easily compared from Figure 5, since the TARC axis of each plot has the same 13 dB range of difference encompassing the lowest and highest TARC value.

To explain why the MIMO antenna with the lowest $|S_{nm}|$ has a minor PDF dispersion, we can refer to (Equation 5). In this equation, there are 16 terms that contribute to the calculated TARC. All these terms are random variables, except $S_{11}$, $S_{21}$, $S_{31}$, and $S_{41}$ that do not contain the random phases. Therefore, the remaining 12 terms are random variables that when added, a new random variable results. The addition of two random variables is equal to the convolution of both random variables, and this convolution always results in a higher PDF dispersion in the domain interval of the resulting random variable. However, when ports exhibit very low mutual coupling, the random variables of the following type:(6)$$\begin{array}{ccc} X=S_{nm}e^{j\theta_{1,m}} & \forall n\neq m, & m\geq 2 \end{array}$$
are especially low, and then Equation (Equation 5) can be approximated to an equation that has only 3 significant random variables of the 12 that are involved in (Equation 5). This necessarily means that the TARC random variable will have less PDF dispersion when the isolation between ports is very high. In the limit, when $S_{nm}\to 0$, the TARC is no longer a random variable and there is null dispersion.

In Figure 5a, the TARC of the MIMO PIFA with the lowest isolation results in an E[TARC] of −10.66 dB. This value is close to −10 dB, and as shown in this figure, the random phases can cause the TARC to increase to −6.9 dB which is the upper limit of the $\Delta CI_{\mathrm{TARC}}$, reducing the bandwidth of the MIMO array for many combinations of the input phases. This is an important observation, since the histogram was taken at 5.82 GHz, which is very close to the resonant frequency where E[TARC] is expected to reach its minimum value. This degradation becomes even more significant as the cut-off frequencies are approached. In Figure 5b,c, the expected value is lower and most of the time the TARC stays below −10 dB, maintaining the bandwidth of MIMO PIFA 2 and MIMO PIFA 3.

Figure 6 presents the probability density distributions for each MIMO antenna at the resonant frequency, where the reflection coefficients are minimum, and the mutual coupling varies for each antenna. From the figure it is observed that as the mutual decoupling increases, the E[TARC] reduces. In Figure 6a the E[TARC] starts at −10.7 dB, it drops to −21.8 dB in Figure 6b, and reaches a value of e29.4 dB in Figure 6c, resulting in a total enhancement of 18.7 dB. This indicates an increase in the performance of the TARC, making the E[TARC] a useful parameter for measuring and comparing the depth of the TARC.

A usual way to present the TARC is through a set of curves associated with several combinations of the phases of the input signals to the MIMO ports, and then the cases where the TARC is degraded above the −10 dB limit are identified. Unlike this traditional approach, this work considers a realistic scenario where any combinations of phases can occur at the input ports of the MIMO antenna. Because we are dealing here with a massive set of combinations of phases, the uncertainty to estimate the effectiveness of a MIMO system is reduced considerably.

Using the same conditions utilized to obtain the results of Figure 5 and Figure 6, the analysis is extended for all measured frequencies from 5.6 to 6.0 GHz. In this particular case, the number of measured frequencies was 401 samples provided by the network analyzer equipment employed to obtain the S-parameters. Once all the calculations have been performed using MATLAB^®^, the TARC graph is obtained. To include the probability information for each frequency, a 3D graphic can be used where the TARC depends on two variables: frequency and probability. This analysis results in the 3D TARC plots of Figure 7 for the three MIMO PIFAs. This representation is useful for measuring bandwidth and visualizing the TARC PDF dispersion. In these plots, the probability variable is mapped onto a rainbow color scale where red indicates a high probability that the TARC has a given value, while blue indicates a low probability. Figure 7a shows that the PDF dispersion of MIMO PIFA 1 extends above −10 dB of TARC, reducing its bandwidth. In contrast with Figure 7b, the PDF dispersion of MIMO PIFA 2 is reduced and more red regions are present, although some blue regions are still seen. Here, there is no bandwidth reduction because the TARC remains below −10 dB. In MIMO PIFA 3 (Figure 7c), there is a noticeable reduction in the PDF dispersion, and the red regions are even larger. Due to the low mutual coupling in MIMO PIFA 3, the TARC behavior is similar to the response of the $S_{nn}$ parameter, thus maintaining the full 200 MHz bandwidth and a value close to −30 dB at the resonant frequency.

For ease of visualization, the probability variable is projected onto the base of the 3D plot as a shadow on a 2D plot, assigning the probability value to the same rainbow color scale. Thus, a TARC value at a given frequency with a high probability of occurrence will be illustrated in red, while a TARC value at the same frequency but with a low probability of occurrence will appear in blue. In this way, the plots in Figure 8 are obtained for the three MIMO antennas. Note that the shape of these plots resembles a 3D TARC shadow, so we can refer to this type of plot as a “TARC shadow”. Along with the TARC shadow, a dashed line representing the expected value obtained at each frequency evaluated is added to the plots in Figure 8. It is important to note that this line of the expected value runs along the high probability zones for the three MIMO antennas studied. Additionally, this 2D TARC representation also facilitates the visualization of the $\Delta CI_{\mathrm{TARC}}$, whose upper and lower limits are represented by the dashed–dotted and dotted lines in Figure 8, respectively.

The TARC shadow demonstrates in a way that leaves no room for doubt if the MIMO antenna is well matched for each random combination of phases of the input signals to the MIMO ports. In the case of MIMO PIFA 1, the TARC shadow, as seen in Figure 8a, is wide due to the high PDF dispersion at all frequencies. For example, at 5.7 GHz there is a low probability, indicated by the blue color at the upper edge of the TARC shadow, that the TARC will increase to −2.4 dB, where the antenna is no longer well matched. Certainly, this TARC value occurs with low probability, but even this low probability is not acceptable in applications where phase management is crucial for good performance, as in the case of beamforming arrays. Even the expected values presented by the dashed curve in Figure 8a display a small range spanning 5.73 GHz to 5.86 GHz, where the TARC is just below −10 dB. However, this range should not be considered a useful bandwidth. A more appropriate bandwidth specification should be one where the entire TARC shadow is kept below −10 dB, and MIMO PIFA 1 does not meet this criterion and therefore has no useful bandwidth at all.

A different case is MIMO PIFA 2 illustrated in Figure 8b, where the TARC shadow is always below the −10 dB limit under all conditions, except for a small proportion of phase combinations where the TARC rises to −9.5 dB near the lower cut-off frequency of 5.7 GHz. For this antenna, the TARC PDF dispersion is also wide, but its value always remains below −10 dB. There are even conditions where the TARC is excellent; for example, at 5.77 GHz, it falls to near −40 dB. However, these conditions are of low probability, as seen from the ample blue zone where the TARC has the lowest values. The limits of the $\Delta CI_{\mathrm{TARC}}$ enclose the green, yellow, and red areas, which have lower PDF dispersion, compared to MIMO PIFA 1.

The best performance is achieved with MIMO PIFA 3 where the PDF dispersion is significantly reduced and the TARC is always below −10 dB within the bandwidth, as seen in the TARC shadow in Figure 8c. In addition, the expected value of −30 dB at the 5.8 GHz resonance frequency is the lowest of the three MIMO antennas studied. These characteristics make MIMO PIFA 3 the best performing antenna, regardless of the random conditions of the propagation channel. The significant reduction in the $\Delta CI_{\mathrm{TARC}}$ for this antenna demonstrates that if $|S_{nm}|\to 0$, the TARC loses its random characteristic and approaches the ideal behavior.

If the bandwidth is maintained under all circumstances, then the efficiency is not affected, and the effectiveness of the MIMO system is always guaranteed. This fact is clearly seen in (Equation 7), which relates the radiation efficiency in multiport antennas with the TARC [9,22,23].(7)$$\eta_{\mathrm{rad}}=1-{\left|\mathrm{TARC}\right|}^{2}$$

If the TARC has small variations, then the radiation efficiency follows these small variations.

In summary, the proposed tools, $\Delta CI_{\mathrm{TARC}}$ and TARC shadow, emerge as very useful tools to assess the characteristics of the TARC random variable, namely, the variability of the TARC under random conditions of the propagation channel, the probability with which a specific value can occur, and the guaranteed bandwidth and radiation efficiency of a MIMO antenna.

## 4. Radiation and MIMO Performance Parameters

Figure 9 displays photographs of the three constructed MIMO PIFAs; the frontal views of MIMO PIFA 1 and 2 are illustrated in Figure 9a and Figure 9b, respectively. The bottom view of MIMO PIFA 3 is seen in Figure 9c. These prototypes use a Taconic TLY-5 laminate with the specifications aforementioned in Section 2 (Figure 1a). In the three photographs, the return connections are visible, implemented using AWG 26 copper vias that pass through the substrate connecting the top plate of each PIFA element to the current return plane. These prototypes were measured to compare and validate their radiation and MIMO performance.

Figure 10 presents the measured and simulated radiation patterns at 5.8 GHz for the three MIMO PIFAs. Since the four elements have the same radiation characteristics, the characterized antenna under test (AUT) was element 1 of each MIMO antenna. At the same time, the other ports were terminated with 50 Ω loads. Each plotted pattern includes the azimuth and elevation cutoff planes.

Overall, Figure 10 displays a high convergence of the measured and simulated radiation patterns, being more noticeable in the main lobe direction. The back direction indicates that the measured and simulated results diverge more significantly, which can be attributed to some error sources, such as the measurement cables, antenna alignment, and reflections inside the semi-anechoic chamber.

For a more complete analysis of the radiation characteristics, a comparison of the co-polarization and cross-polarization components in both the azimuth and elevation planes for the three MIMO antennas is included in Figure 11. According to what is observed, there is also a close agreement of the measured and simulated patterns.

On the other hand, Figure 12 compares the measured and simulated realized gains and includes the simulated radiation efficiency. The absolute realized gain (which includes the $G_{\theta }$ and $G_{\varphi }$ components) is presented for each MIMO antenna. The realized gain is taken from the best value in the azimuthal plane because this plane contains the maximum gain value, as shown in Figure 12.

From Figure 12, the peak gain can be obtained. MIMO PIFA 1 has a peak gain of 6.22 dB at 5.75 GHz, while MIMO PIFA 2 has a peak gain of 6.73 dB at 5.83 GHz. Within the bandwidth of interest, from 5.7 GHz to 5.9 GHz, the realized gain is always higher than 5.47 dB and 6.5 dB for MIMO PIFA 1 and 2, respectively. With respect to MIMO PIFA 3, this has a peak gain of 7.16 dB at 5.77 GHz and over the entire bandwidth of interest maintains its gain above 6.62 dB, which is the best performance of the three MIMO antennas.

The gain curves of each MIMO antenna are similar to each other, as well as their radiation efficiency curves, which are always above 98% over the entire bandwidth, as illustrated in Figure 12. This means that the changes made to the current return connection and the introduction of the DGS in MIMO PIFA 3 do not significantly affect gain and efficiency, and more importantly, they do improve the elements’ decoupling performance.

The envelope correlation coefficient (ECC or $\rho_{e_{nm}}$) is an essential MIMO parameter that evaluates the independence of two radiators in a multipath environment [24]. When the propagation environment is isotropic and uniform, the ECC between ports *n* and *m* is given by [25]:(8)$$\rho_{e_{nm}}=\frac{{\left|\underset{4\pi }{\;\int \int \;}\left[\vec{F_{n}}\left(\theta ,\varphi \right)•\vec{F_{m}}\left(\theta ,\varphi \right)\right]d\Omega \right|}^{2}}{\underset{4\pi }{\;\int \int \;}{\left|\vec{F_{n}}\left(\theta ,\varphi \right)\right|}^{2}d\Omega \underset{4\pi }{\;\int \int \;}{\left|\vec{F_{m}}\left(\theta ,\varphi \right)\right|}^{2}d\Omega }$$
where • denotes the Hermitian product, $F_{n}\left(\theta ,\varphi \right)$ and $F_{m}\left(\theta ,\varphi \right)$ are the 3D radiation patterns of the *n*th and *m*th elements of the MIMO antenna, and dΩ is the solid angle differential.

The ECC can also be approximated using a simpler expression that only applies for antennas with high radiation efficiency (greater than 90%) and uses only the complex S-parameters, as reported in [25]:(9)$$\rho_{e_{nm}}=\frac{{\left|S_{nn}^{*}S_{nm}+S_{mn}^{*}S_{mm}\right|}^{2}}{\left[\prod_{k=n,m}\left(1-\sum_{l=n,m}{\left|S_{lk}\right|}^{2}\right)\right]}$$

In this paper, the simulated ECC is calculated using proprietary software [26] that employs Equation (Equation 8) and the antenna radiation patterns ($F_{n,m}\left(\theta ,\varphi \right)$) provided by CST. Since all three MIMO antennas meet the 90% radiation efficiency criterion, the measured ECC is also calculated using Equation (Equation 9) and the measured S-parameters. The results of (Equation 8) and (Equation 9) are plotted and compared in Figure 13.

In general, the simulated and measured ECC curves shown in Figure 13 follow the same trend. In Figure 13b, the dashed horizontal-colored lines specify the maximum measured ECC for each of the three MIMO antenna within the bandwidth of interest. Comparing these results, MIMO PIFA 1 has the highest ECC in the entire bandwidth of interest, which is 0.01 at 5.7 GHz concerning ports 1 and 2. Note that the scale of the plots corresponding to the ECC curve for MIMO PIFA 1 is different from that of the other two MIMO antennas because its ECC is significantly higher and the scale on the right of each plot in Figure 13 is different from the scale on the left.

As for MIMO PIFA 2 and 3, the measured correlation is very low. In MIMO PIFA 2, the maximum measured ECC drops to 0.00046 at 5.7 GHz, which occurs between adjacent ports 1 and 2, whereas in MIMO PIFA 3, the maximum measured ECC between ports 1 and 2 is even lower, 0.0003 at 5.9 GHz. These results demonstrate that the reduction in mutual coupling increases the ports’ independence in a MIMO antenna. In all three MIMO antennas, the measured ECC concerning adjacent ports 1 and 3 is close to that found with ports 1 and 2. Meanwhile, the ECC for the opposite pair of ports 1 and 4 is well below the ECC for adjacent ports. With such low ECC values, maximum MIMO performance is guaranteed with respect to the radiation characteristics of each MIMO antenna element.

Related to ECC, diversity gain (DG) is also an important parameter for MIMO antennas. The diversity gain quantifies the decrease in SNR of a MIMO system compared to a non-diversity system [10,24]. The DG between ports *n* and *m* is calculated by:(10)$$DG=10\sqrt{1-\rho_{e_{nm}}}$$

Table 1 presents the DG for the maximum measured ECC of each MIMO antenna. As expected, the DG improves as the ECC does, with MIMO PIFA 3 having the best DG, which is closer to 10 concerning ports 1 and 2.

Another important MIMO parameter is the capacity loss ($C_{\mathrm{loss}}$), which determines the losses of the channel capacity in a real MIMO system where the correlation coefficient is different from zero. The expression to evaluate the capacity loss is the following [8]:(11)$$C_{\mathrm{loss}}=-\log_{2}\left(\det\Psi^{R}\right)$$
where $\Psi^{R}$ is the correlation matrix of the receiving antenna. This matrix can be easily obtained for two-element MIMO antennas [8,27], but the complexity increases as the number of elements grows. For the studied case of a four-element MIMO antenna, the correlation matrix is [28,29]:(12)$$\Psi^{R}=\left(\begin{array}{cccc} \rho_{11} & \rho_{12} & \rho_{13} & \rho_{14}\\ \rho_{21} & \rho_{22} & \rho_{23} & \rho_{24}\\ \rho_{31} & \rho_{32} & \rho_{33} & \rho_{34}\\ \rho_{41} & \rho_{42} & \rho_{43} & \rho_{44} \end{array}\right)$$
where:(13)$$\rho_{nm}=\left\{\begin{array}{cc} 1-\left|\sum_{k=1}^{4}S_{nk}^{*}S_{kn}\right|, & \mathrm{if}\;n=m\\ -\left|\sum_{k=1}^{4}S_{nk}^{*}S_{km}\right|, & \mathrm{if}\;n\neq m \end{array}\right.$$

Using Equations (Equation 11)–(Equation 13) and the measured S-parameters, the $C_{\mathrm{loss}}$ is calculated and the results are depicted in Figure 14 for the three MIMO antennas. As seen in this figure, MIMO PIFA 1 has the highest $C_{\mathrm{loss}}$ over the entire bandwidth. This result confirms that, like ECC, $C_{\mathrm{loss}}$ also depends on mutual coupling.

## 5. Comparison with Other Works

The MIMO PIFA antenna with DGS is compared in Table 2 with other single-band MIMO PIFA antennas reported in the literature. As observed from the table, the MIMO antenna with DGS achieves the highest isolation, and this is accomplished with a small separation of 0.5$\lambda_{0}$ among adjacent ports. As described in this paper, this high isolation was obtained using the polarization diversity technique, optimizing the position of the current return connection of every PIFA element and introducing a cross-shaped slot that is not demanding extra space in the footprint of the MIMO antenna. Additionally, the proposed MIMO antenna with DGS has higher gain and a very low ECC, which is indicative of the radiation patterns being largely independent. In this work, the TARC is the lowest of Table 2, although only two of the other antennas reported this parameter. In Table 3 more details of the TARC are given, specifically, the range of TARC values for the conditions when $|S_{nm}\left|\;<\;\right|S_{nn}|$, $|S_{nm}\left|\;\approx \;\right|S_{nn}|$, and $|S_{nm}\left|\;>\;\right|S_{nn}|$.

## 6. Conclusions

In this paper, the TARC parameter of a MIMO antenna has been extensively studied by means of the statistical analysis of a very large set of random phases of the signals arriving at the input ports of a MIMO antenna. This large number of phase combinations allows us to emulate the real propagation channel conditions under which a MIMO antenna must operate with the best possible performance.

A visual representation of the TARC parameter, named here as “TARC shadow”, has been derived from the statistical analysis to determine unambiguously and at first glance whether a MIMO antenna preserves its bandwidth under all possible random conditions of the input signals and thus whether the radiation efficiency is not compromised for particular random conditions of the communication channel. The introduced concept of “TARC shadow” accurately describes the PDF dispersion and probability of the TARC values for any selected frequency.

Since the “TARC shadow” PDF dispersion depends on the mutual coupling and the convolution of the random variables determined by the input phases, the relationship between “TARC shadow” and various mutual coupling conditions was investigated in three MIMO PIFAs. The study proved that for a low mutual coupling, with values lower than the reflection coefficients, the “TARC shadow” PDF dispersion diminishes. This indicates that the MIMO antenna enhances robustness against random phases variations and maintains an adequate performance under all conditions of the communication channel. In the scenario in which the mutual coupling is close to zero, the TARC becomes quasi-independent of the random phases. In this case, the TARC calculation mainly depends on the reflection coefficients. These results are of significant importance to improve the accuracy of the propagation channel models because they fully consider the effect of the input phases that impacts the efficiency of MIMO systems and multiport antennas.

## Figures and Tables

**Figure 1 sensors-25-04171-f001:**
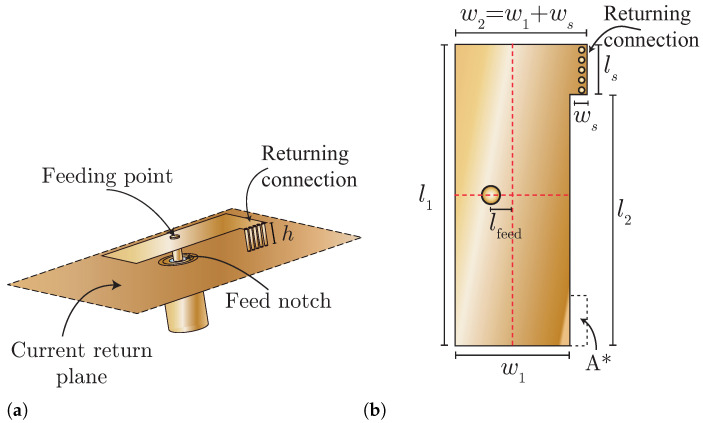
Geometry and dimensions of the proposed PIFA element. (**a**) Perspective view of the PIFA. (**b**) Dimensions of the PIFA.

**Figure 2 sensors-25-04171-f002:**
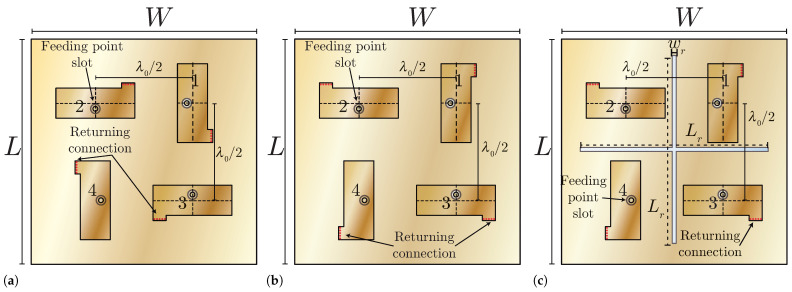
Proposed MIMO arrays for the TARC analysis. (**a**) MIMO PIFA 1. (**b**) MIMO PIFA 2. (**c**) MIMO PIFA 3.

**Figure 3 sensors-25-04171-f003:**
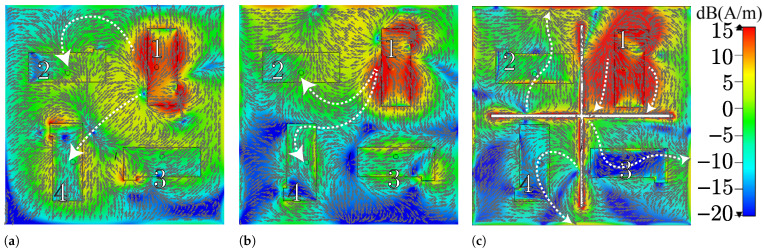
Current distributions on the return plane. (**a**) MIMO PIFA 1. (**b**) MIMO PIFA 2. (**c**) MIMO PIFA 3.

**Figure 4 sensors-25-04171-f004:**
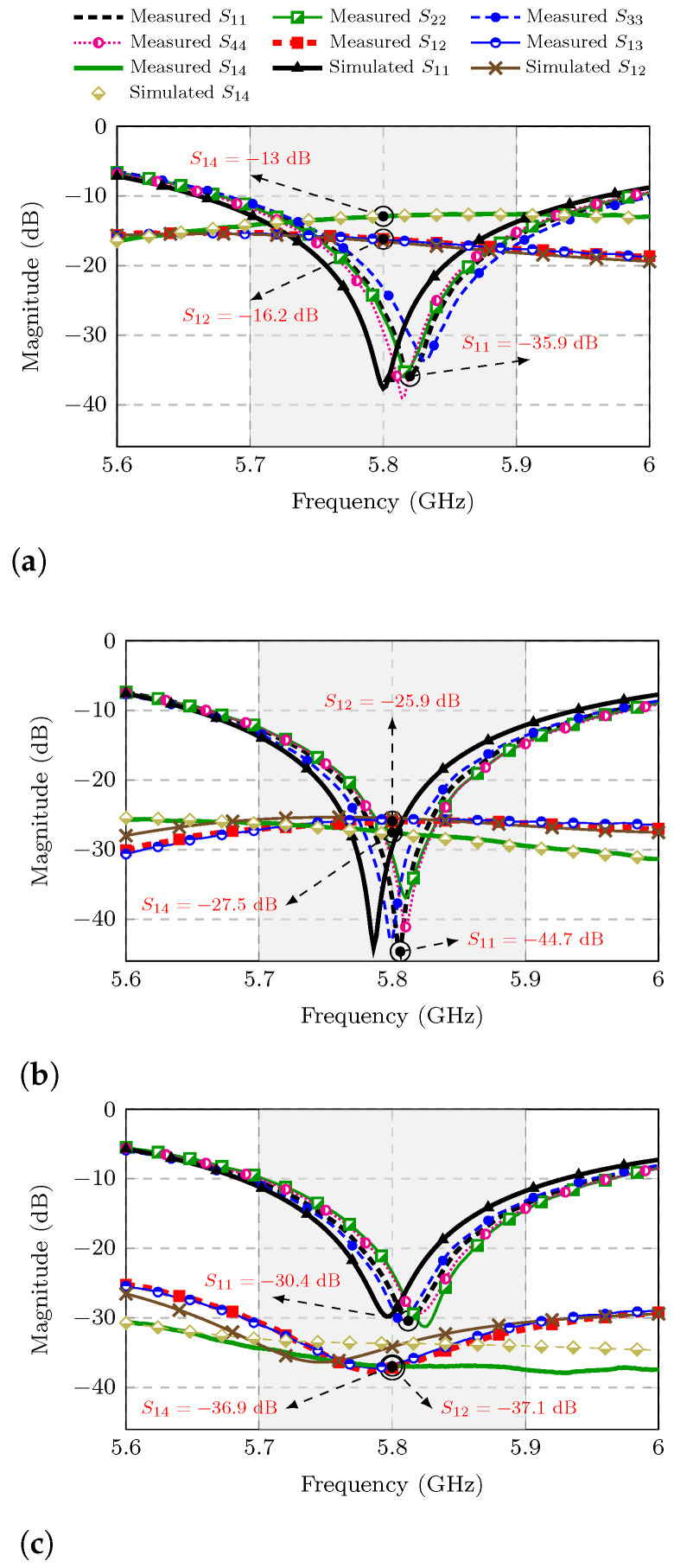
Simulated and measured S-parameters. (**a**) MIMO PIFA 1. (**b**) MIMO PIFA 2. (**c**) MIMO PIFA 3.

**Figure 5 sensors-25-04171-f005:**
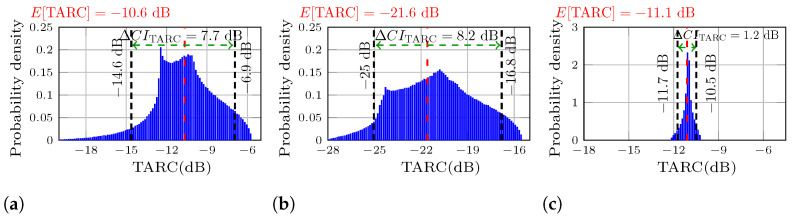
Probability distributions of the calculated TARC using the measured S-parameters. (**a**) MIMO PIFA 1, best case for the condition $|S_{nm}\left|\;>\;\right|S_{nn}|$ at 5.82 GHz. (**b**) MIMO PIFA 2, best case for the condition $|S_{nn}\left|\;\approx \;\right|S_{nm}|$, at 5.78 GHz. (**c**) MIMO PIFA 3, best case for the condition $|S_{nm}\left|\;<\;\right|S_{nn}|$, at 5.7 GHz.

**Figure 6 sensors-25-04171-f006:**
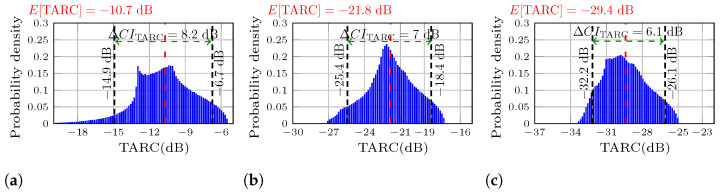
Probability density of the calculated TARC using the measured S-parameters at the resonant frequency. (**a**) MIMO PIFA 1. (**b**) MIMO PIFA 2. (**c**) MIMO PIFA 3.

**Figure 7 sensors-25-04171-f007:**
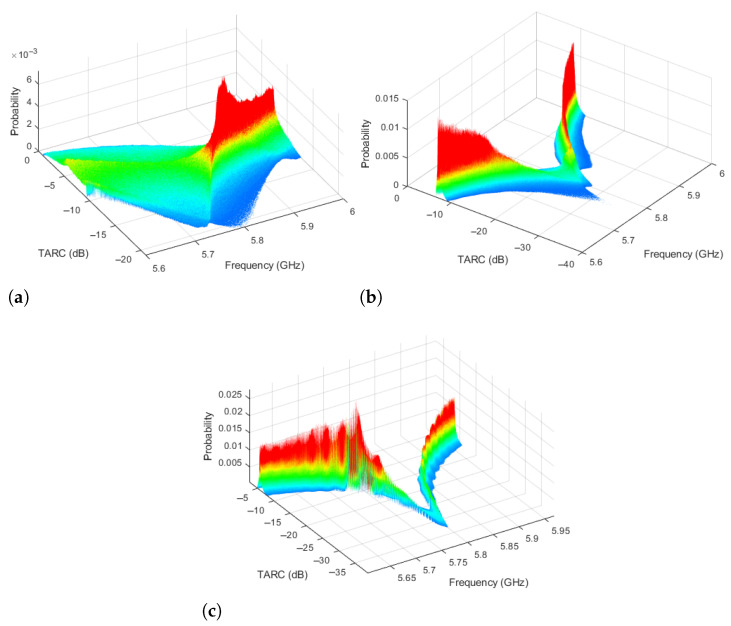
Three-dimensional probability TARC. (**a**) MIMO PIFA 1. (**b**) MIMO PIFA 2. (**c**) MIMO PIFA 3.

**Figure 8 sensors-25-04171-f008:**
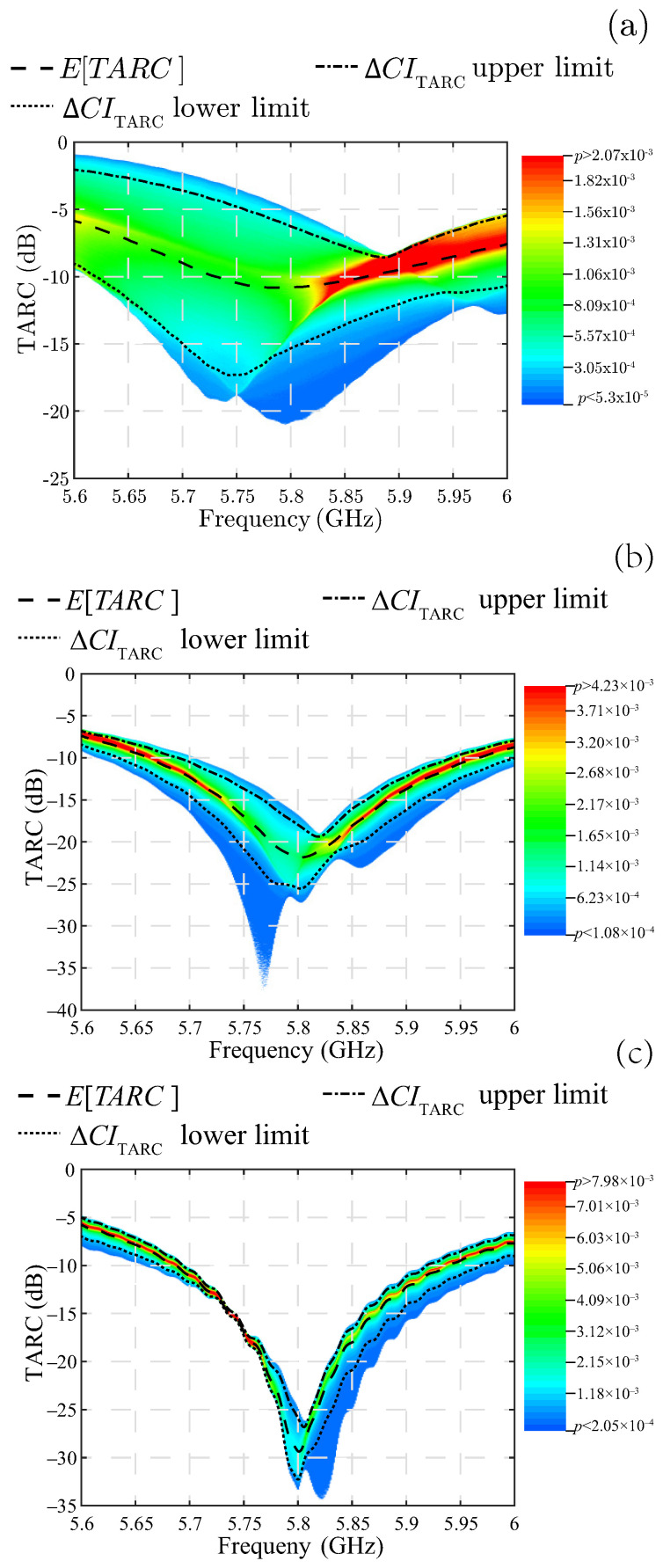
TARC shadow. (**a**) MIMO PIFA 1. (**b**) MIMO PIFA 2. (**c**) MIMO PIFA 3.

**Figure 9 sensors-25-04171-f009:**
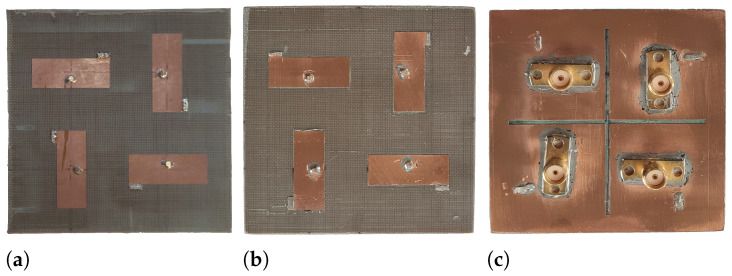
Photography of the developed MIMO PIFAs. (**a**) MIMO PIFA 1. (**b**) MIMO PIFA 2. (**c**) MIMO PIFA 3.

**Figure 10 sensors-25-04171-f010:**
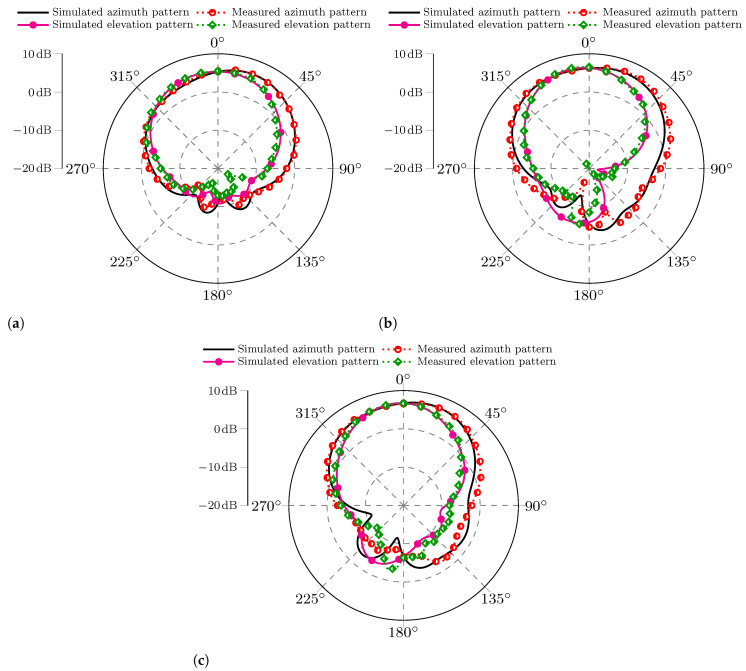
Comparison of the measured and simulated radiation patterns for azimuth and elevation planes. (**a**) MIMO PIFA 1. (**b**) MIMO PIFA 2. (**c**) MIMO PIFA 3.

**Figure 11 sensors-25-04171-f011:**
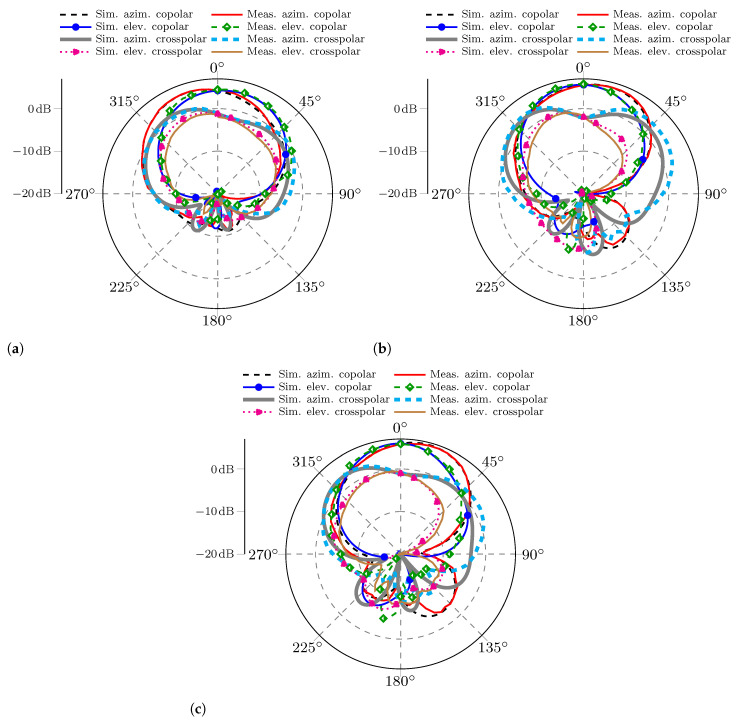
Comparison of the measured and simulated azimuth and elevation planes; the components presented are the co-polar and cross-polar components for the three MIMO PIFAs. (**a**) MIMO PIFA 1. (**b**) MIMO PIFA 2. (**c**) MIMO PIFA 3.

**Figure 12 sensors-25-04171-f012:**
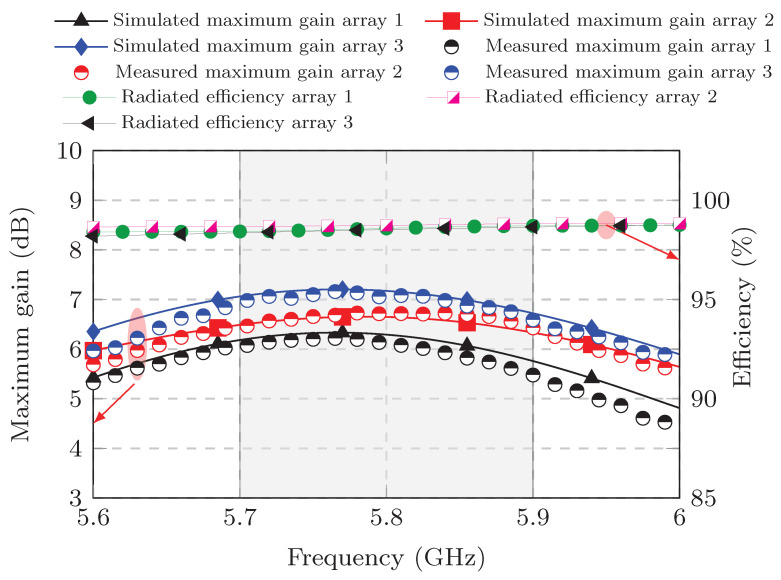
Results of the realized gain and radiated efficiency of the PIFA element 1 of each array.

**Figure 13 sensors-25-04171-f013:**
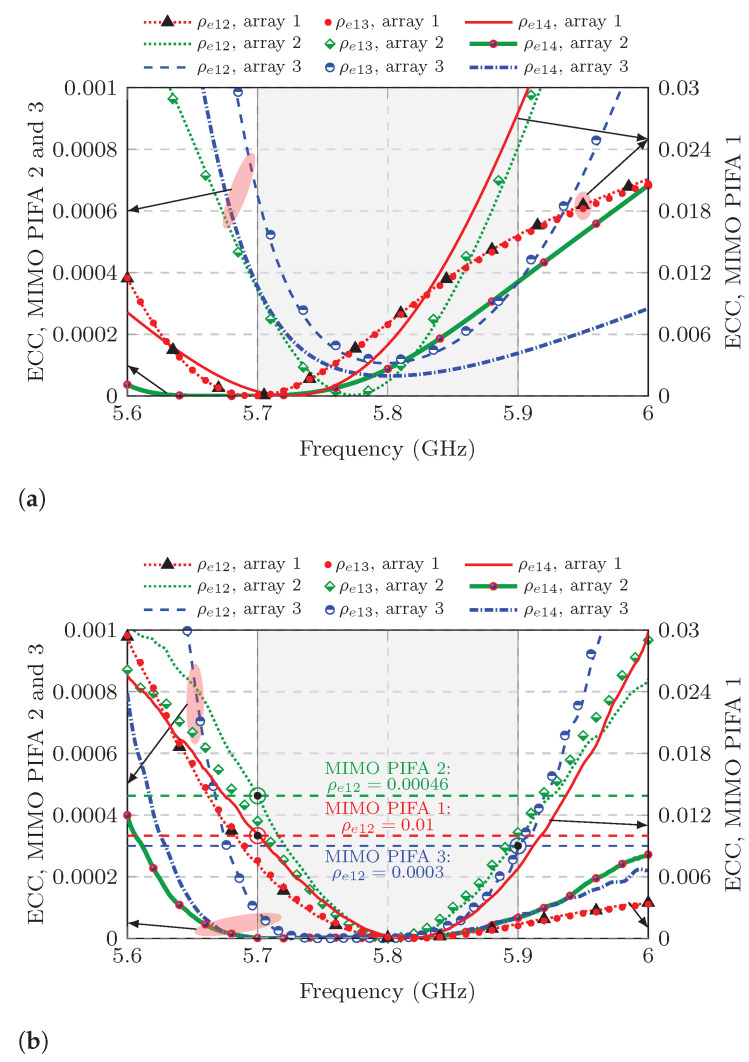
ECC of the studied MIMO PIFAs. (**a**) Simulated ECC. (**b**) Measured ECC.

**Figure 14 sensors-25-04171-f014:**
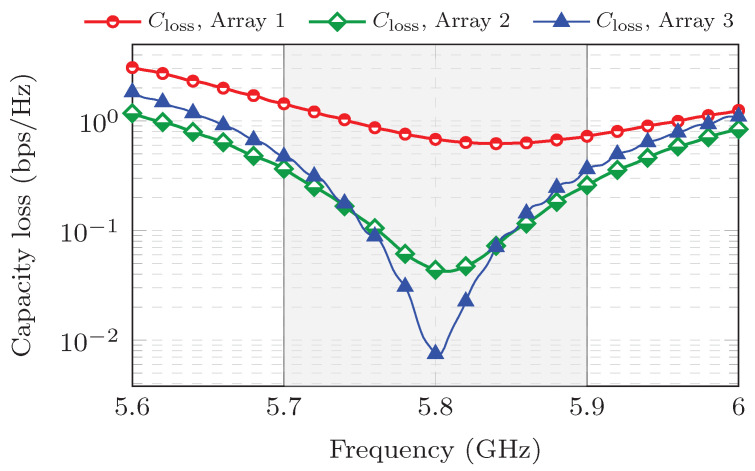
Measured capacity loss for the MIMO PIFAs.

**Table 1 sensors-25-04171-t001:** Threshold of diversity gain for the three proposed MIMO PIFAs.

MIMO PIFA	DG	Frequency	Ports
1	9.9498	5.7 GHz	1 and 2
2	9.9769	5.7 GHz	1 and 2
3	9.9849	5.9 GHz	1 and 2

**Table 2 sensors-25-04171-t002:** Comparison of the performance of the proposed PIFA-MIMO antenna with other single-band PIFA-MIMO antennas.

Reference	Number of Ports	Size	Minimum Ports Separation	Fractional Bandwidth	Isolation	Gain	Efficiency	ECC	TARC
[30]	4	$0.5\lambda_{0}\times \lambda_{0}$	$1.12\lambda_{0}$	30%	>10 dB	4.5 dB	${80\%}^{\;b}$	<$81\times 10^{-6}$	NR
[31]	2	$0.9\lambda_{0}\times 0.36\lambda_{0}$	$0.25\lambda_{0}$	35%	>10 dB	NR	NR	<0.02	NR
[32]	2	$0.94\lambda_{0}\times 1.88\lambda_{0}$	$0.94\lambda_{0}$	14.2%	>13 dB	6.57 dB	${75\%}^{\;b}$	<0.032	<−8 dB
[33]	2	$0.9\lambda_{0}\times 1.8\lambda_{0}$	$0.72\lambda_{0}$	20%	>23.5 dB	NR	NR	NR	NR
[34]	2	$0.79\lambda_{0}\times 1.57\lambda_{0}$	$0.79\lambda_{0}$	30.3%	>12 dB	4 dB	${72\%}^{\;b}$	<0.01	<−7 dB
[35]	3	$0.96\lambda_{0}\times 1.89\lambda_{0}$	$0.5\lambda_{0}$	2.9%	>13 dB	6.06 dBi	${97\%}^{\;a}$	<0.0051	NR
[36]	2	$0.16\lambda_{0}\times 0.6\lambda_{0}$	$0.13\lambda_{0}$	4%	>22 dB	1.62 dBi	${90\%}^{\;a}$	<0.0035	NR
[37]	3	$\lambda_{0}\times 2.3\lambda_{0}$	$1.3\lambda_{0}$	5%	>29 dB	5.35 dB	NR	NR	NR
[38]	8	$0.88\lambda_{0}\times 1.75\lambda_{0}$	$0.14\lambda_{0}$	3%	>17 dB	5.2 dBi	${58\%}^{\;b}$	<0.3	NR
[39]	1	$0.38\lambda_{0}\times 0.31\lambda_{0}$	NA	220%	NA	2.1 dBi	${90\%}^{\;a}$	NA	NA
[40]	1	$0.75\lambda_{0}\times 0.7\lambda_{0}$	NA	11%	NA	4.1 dBi	${95\%}^{\;a}$	NA	NA
This work	4	$1.16\lambda_{0}\times 1.16\lambda_{0}$	$0.5\lambda_{0}$	3.4%	>31.5 dB	7.16 dB	${98\%}^{\;b}$	<$2.9\times 10^{-4}$	<−11.7 dB

NR: Not reported, NA: not applicable, ^a^ radiation efficiency, and ^b^ total efficiency.

**Table 3 sensors-25-04171-t003:** Comparison of the TARC of the proposed PIFA-MIMO antenna with the aforementioned works.

Reference	TARC When $|S_{\mathrm{nm}}\left|\;<\;\right|S_{\mathrm{nn}}|$ (Best Case)	TARC When $|S_{\mathrm{nm}}\left|\;\approx \;\right|S_{\mathrm{nn}}|$ (Best Case)	TARC When $|S_{\mathrm{nm}}\left|\;>\;\right|S_{\mathrm{nn}}|$ (Best Case)	Phase Probability Distribution
[30]	NR	NR	NR	NR
[31]	NR	NR	NR	NR
[32]	−11.7 dB to −18.6 dB	−17 dB to −30.4 dB	−18.6 dB to −24 dB	Uniform
[33]	NR	NR	NR	NR
[34]	−7.1 dB	−13 dB	−16.4 dB	NA
[35]	NR	NR	NR	NR
[36]	NR	NR	NR	NR
[37]	NR	NR	NR	NR
[38]	NR	NR	NR	NR
[39]	NA	NA	NA	NA
[40]	NA	NA	NA	NA
This work	−10.2 dB to −12.8 dB	−25.3 dB to −33.5 dB	There are no cases	Uniform

NR: not reported, NA: not applicable.

## Data Availability

All insights and original data analyses from this research are included in the article. For additional information, please contact the corresponding author.

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
