# Peer review of "An In-Depth Statistical Analysis of the TARC Parameter to Evaluate the Real Impact of Random Phases in MIMO Antennas"

_sensors, 2025, doi:10.3390/s25134171_

Round 1

Reviewer 1 Report

Comments and Suggestions for Authors

The authors present a statistical analysis of the total active reflection coefficient with three four-port MIMO antennas. Results indicate its good performance. But some problems still need to be further discussed:

  1. In Eq. (4), is the random phase difference of ports 2-4 the same as that of port 1? Usually, the phase difference between different ports varies, so what is the principle by performing this?
  2. In Figs. 5 and 6, the probability distributions and density of the measured TARC are presented, but how is the random phase difference set in the measurement?
  3. It seems that Section 4 which exhibits radiation and MIMO performance parameters is not that related to the proposed TARC, but to exhibit the antenna performance with low mutual coupling. So what is the intention of this section? And other parameters which can support the TARC improvement should be included.

Author Response

The authors appreciate the comments of reviewer 1, we include a file with the reply to the comments adding your suggestions to the manuscript to improve  the clarity of its content.

Reviewer 2 Report

Comments and Suggestions for Authors

Comments on the Quality of English Language

The manuscript is generally well-written and the technical content is clearly communicated. 

Author Response

The authors appreciate the comments of reviewer 2, we include a file with the reply to the comments adding your suggestions to the manuscript to improve  the clarity of its content.

Reviewer 3 Report

Comments and Suggestions for Authors

This manuscript presents a statistical analysis of the Total Active Reflection Coefficient (TARC) under random phase excitations in MIMO antennas. The introduction of the “TARC shadow” visualization is interesting, and the study is supported by both simulation and measurement. However, several key issues should be addressed:

1. The analysis is based solely on ideal S-parameters and random phases, without considering fabrication tolerances, feedline asymmetries, or environmental effects. This may underestimate performance variability in real systems.

2. Although the authors mention MATLAB-based simulations, no code, algorithm description, or pseudocode is provided.

3. Figure 10 lacks cross-polarization components, which are important for evaluating polarization purity and MIMO performance.

4. The manuscript lacks a comparison with other 5.8 GHz MIMO antennas, such as doi: 10.1109/ACCESS.2023.3333881 and doi: 10.1017/S1759078720001099.

5. The “TARC shadow” concept is not clearly differentiated from existing methods such as TARC envelopes or statistical confidence bands. A clearer explanation of its novelty is recommended.

Author Response

The authors appreciate the comments of reviewer 3, we include a file with the reply to the comments adding your suggestions to the manuscript to improve  the clarity of its content.

Round 2

Reviewer 1 Report

Comments and Suggestions for Authors

I have no more comment.

Reviewer 3 Report

Comments and Suggestions for Authors

This manuscript has been effectively revised and is now suitable for acceptance.